# Interleukin-4 Promotes Tuft Cell Differentiation and Acetylcholine Production in Intestinal Organoids of Non-Human Primate

**DOI:** 10.3390/ijms22157921

**Published:** 2021-07-24

**Authors:** Akihiko Inaba, Ayane Arinaga, Keisuke Tanaka, Takaho Endo, Norihito Hayatsu, Yasushi Okazaki, Takumi Yamane, Yuichi Oishi, Hiroo Imai, Ken Iwatsuki

**Affiliations:** 1Department of Nutritional Science and Food Safety, Faculty of Applied Bioscience, Tokyo University of Agriculture, Tokyo 156-8502, Japan; inaba.akihiko.74a@st.kyoto-u.ac.jp (A.I.); arina.ayane12@gmail.com (A.A.); ty204887@nodai.ac.jp (T.Y.); y3oishi@nodai.ac.jp (Y.O.); 2Molecular Biology Section, Department of Cellular and Molecular Biology, Primate Research Institute, Kyoto University, Kyoto 484-8506, Japan; 3Genome Research Center, Tokyo University of Agriculture, Tokyo 156-8502, Japan; kt205453@nodai.ac.jp; 4RIKEN Center for Integrative Medical Sciences, Yokohama 230-0045, Japan; takaho.endo@riken.jp (T.E.); norihito.hayatsu@riken.jp (N.H.); yasushi.okazaki@riken.jp (Y.O.)

**Keywords:** tuft cell, acetylcholine, intestine, organoid, primate, IL-4

## Abstract

In the intestine, the innate immune system excludes harmful substances and invading microorganisms. Tuft cells are taste-like chemosensory cells found in the intestinal epithelium involved in the activation of group 2 innate lymphoid cells (ILC2). Although tuft cells in other tissues secrete the neurotransmitter acetylcholine (ACh), their function in the gut remains poorly understood. In this study, we investigated changes in the expression of genes and cell differentiation of the intestinal epithelium by stimulation with interleukin-4 (IL-4) or IL-13 in macaque intestinal organoids. Transcriptome analysis showed that tuft cell marker genes were highly expressed in the IL-4- and IL-13-treated groups compared with the control, and the gene expression of choline acetyltransferase (ChAT), a synthesis enzyme of ACh, was upregulated in IL-4- and IL-13-treated groups. ACh accumulation was observed in IL-4-induced organoids using high-performance liquid chromatography-mass spectrometry (HPLC/MS), and ACh strongly released granules from Paneth cells. This study is the first to demonstrate ACh upregulation by IL-4 induction in primates, suggesting that IL-4 plays a role in Paneth cell granule secretion via paracrine stimulation.

## 1. Introduction

The intestinal epithelium is exposed to a variety of chemicals or microorganisms that can be both beneficial and harmful to the body. Thus, to maintain homeostasis, intestinal epithelial cells (IECs) exert multiple sensing mechanisms against nutrients, tastants, microorganisms, and chemicals. The tuft cell (or brush cell) is an atypical epithelial cell-type with a characteristic apical tuft of microvilli, and is found in a variety of organs, including the intestine. It is considered to be a taste-like chemosensory cell because these cells express type 2 taste cell markers, such as POU domain class 2 transcription factor 3 (POU2F3), alpha-gustducin (α-gust or GNAT3), and transient receptor potential cation channel subfamily M member 5 (TRPM5) [1,2]. Recent studies have shown that intestinal tuft cells activate group 2 innate lymphoid cells (ILC2) by secreting interleukin-25 (IL-25), followed by parasite infection [3,4,5]. Activated ILC2 secretes Th2 cytokines, such as IL-13 and IL-4, and promotes the hyperplasia of both tuft and goblet cells, leading to the mucous secretion and activation of intestinal motility, known as the “weep and sweep” response [5]. Other physiological functions of tuft cells in the gut are also proposed for functions such as to sense bitter compounds and succinate that will guide hyperplasia of tuft cells [6,7,8,9].

Although studies using a rodent model have shown that intestinal tuft cells trigger type 2 immune responses in the gut, whether similar functions exist in primates has not yet been investigated. Furthermore, biosynthetic enzymes of leukotrienes, prostaglandins, and acetylcholine (ACh) have been reported to be expressed in human intestinal epithelial cells [10]. However, the mechanism by which these molecules are released remains unclear. Choline acetyltransferase (ChAT), a synthetic enzyme of Ach, is a key enzyme for ACh production and is often used as a tuft cell marker in the body. While it has been reported that tuft cells in the trachea and airways release ACh by noxious or bitter stimuli to exclude these hazardous chemicals [11], the function of ACh from intestinal tuft cells remains elusive.

Recently, we reported on the generation of intestinal organoids from macaques to analyze the intestinal chemosensory cells of primates in vitro [12]. Since macaque organoids are able to differentiate into tuft cells by Th2 cytokines, this system provides a suitable model to explore the immune response of the intestinal epithelium in vitro. Therefore, in the present study, we investigated the response of the intestinal epithelium to Th2 cytokines using a macaque organoid system. Our data indicate that ACh accumulation in tuft cells correlates with the upregulation of ChAT transcripts triggered by Th2 cytokines. Furthermore, we propose that ACh secreted by tuft cells acts in a paracrine manner to positively regulate granule secretion from Paneth cells.

## 2. Results

### 2.1. Transcriptome Analysis of Macaque Intestinal Organoids after 72 h Culture in Differentiation Medium

We tested whether modified media could be used to culture macaque intestinal organoids. The organoids were maintained with the improved media using IGF-1 and FGF-2 (IF medium) (Figure 1). As a result, morphological changes were observed compared with cultures using the conventional medium with p38i (Appendix A). We observed that the conventional method using p38i had a lower proliferation potential than the modified media using IGF-1 and FGF-2, especially when organoids were passaged (Appendix A). Next, we examined whether cell differentiation could be induced by differentiation (DI) medium or DI medium with IL-4 (IL-4 medium), as described in the Materials and Methods (Figure 1a,b). Immunofluorescent studies showed that doublecortin-like kinase 1 (DCLK1) and serotonin (5HT), which are tuft cells and enterochromaffin cell markers [13,14,15], respectively, were rarely detected in IF medium, while the DI medium or IL-4 medium upregulated the expression of both (Appendix A). IL-4 medium had a significant effect on DCLK1 expression (Figure 1d, Appendix A). As the efficiency of cell differentiation into tuft cells by IL-4 induction was maintained at the same level as in our previous report (Appendix A) [12], the IF conditions were used for subsequent experiments.

To observe transcriptional changes under three different culture conditions (IF, DI, and DI with IL-4), we performed RNA-Seq analysis. Principal component analysis (PCA) showed that biological replicates from the same culture conditions fell into the same cluster, indicating that clusters represented distinct groups (Figure 1c).

Using volcano plot analysis, we identified 1197 genes that were significantly upregulated and 734 genes that were downregulated by IL-4 (FC ≥ |1.5| and FDR ≤ 0.01) (Appendix A). The representative genes within the upregulated genes were *Mucin 2* (*MUC2), Trefoil Factor 3 (TFF3), Defensin Alpha 6 (DEFA6), Lysozyme (LYZ), Arachidonate 5-Lipoxygenase (ALOX5)*, and *CHAT*. *L**eucine-rich repeat G-protein-coupled receptor* (*LGR5)* was identified as one of the downregulated genes. Next, to gain further insights into the transcriptional changes in each group, transcriptome data were subjected to heatmap analysis. We compared the gene expression patterns of 88 genes reported as tuft cell markers (Appendix A) [16]. As a result, 30 genes, including *POU2F3, TRPM5, DCLK1, Phospholipase C-beta-2*
*(PLCb2), Phospholipase C-gamma-2*
*(PLCg2), Interleukin-17 receptor B (IL17RB), ALOX5*, and *CHAT**,* were found to be abundantly expressed under IL-4-induced conditions compared with IF or DI conditions (Figure 1d). In contrast, we failed to detect *GNAT3* and *IL-25*, which are tuft cell markers in mouse [17], owing to their low transcript levels.

To substantiate transcriptional regulation by Th2 cytokines, IL-13 was also added to the culture medium, and the expression of tuft cell marker proteins and transcripts was analyzed by immunohistochemistry and RNA-Seq. The results showed that both protein and gene expression patterns were similar to those stimulated by IL-4, indicating that IL-13 also induces tuft cell differentiation (Appendix A).

### 2.2. Confirmation of RNA-Seq Results by Semi-Quantitative RT-PCR

To confirm the results of RNA-Seq, we performed a semi-quantitative reverse transcription polymerase chain reaction (RT-PCR). The expression patterns of tuft cell markers (*DCLK1*, *POU2F3*, *CHAT*, and *PLCb2*) were comparable to the results obtained from RNA-Seq (Figure 2a–d). The expression of a goblet cell marker (*MUC2*) and Paneth cell markers (*LYZ* and *DEFA6*) was significantly increased by IL-4 induction, while that of a stem cell marker (*LGR5*) was decreased (Figure 2e–h). *MUC2*- and *DEFA6*-positive cells were also increased in organoids induced by IL-4 (Appendix A). However, we failed to detect any change in the expression of the enterocyte marker (*Solute Carrier Family 5 Member 1*; *SLC5A1)* between the DI and IL-4 groups (Figure 2i). Although the expression of enteroendocrine cell marker (*Chromogranin A*; *CHGA*) appeared to increase in the IL-4 group (Figure 2j), this change was marginal in the RNA-Seq analysis (data not shown).

### 2.3. ChAT and Ach Are Upregulated by IL-4

RNA-Seq analysis revealed that *ChAT* expression was upregulated when induced by IL-4 (Figure 3a). Immunohistochemical analysis using macaque intestinal tissues and organoid cultures showed that ChAT and DCLK1 were co-expressed in tuft cells (Figure 3b). ChAT immunoreactivity was not observed in the 5HT-positive enteroendocrine cells (Appendix A). As ChAT is a key enzyme responsible for the biosynthesis of ACh [18], we speculated that macaque organoids synthesize ACh. Therefore, we attempted to quantify ACh in organoids by HPLC/MS analysis (Figure 3c). The ACh concentration was evaluated from the standard curve of d4-ACh, as described in the Materials and Methods. From the standard curve, the ACh concentration range was 2.27–183.50 nM (0.4 to 33.3 ppb of ACh-Cl). We also measured the ACh concentration of cytosolic samples prepared from the organoids of control (IF), differentiation (DI), or IL-4 induced (IL-4) conditions, and found that the average concentration of ACh in each culture condition was 7.58, 15.18, and 100.45 nM, respectively. The average value (*N* = 3) of ACh concentration was statistically analyzed, and a bar graph was generated to show that ACh was significantly up-regulated in IL-4-induced conditions compared with the other conditions (Figure 3d).

### 2.4. Granule Secretion Induced by ACh in Organoid Cultures

We observed ACh accumulation in the intestinal organoids of macaques when tuft cell differentiation was induced by IL-4. A previous study showed that Paneth cells express muscarinic ACh receptors [19]. Upon carbachol stimulation, Paneth cells release antimicrobial substances [20]. As carbachol is a synthetic agonist of the cholinergic receptor, we speculated that ACh could also stimulate granule secretion from Paneth cells. Thus, we investigated the effect of ACh on Paneth cells using mouse organoids with distinguishable vesicles in Paneth cells. Upon the basolateral stimulation of mouse intestinal organoids with 10 µM ACh, we detected the release of vesicular contents from Paneth cells (Figure 4a and Appendix A). This phenomenon lasted for 5 min, until the area of granules in Paneth cells was significantly decreased compared with the control PBS stimulation (Figure 4b).

## 3. Discussion

We have recently reported on the generation of intestinal organoids from macaques [12] by modifying the original culture method of the human intestine, as described by Sato et al. [21]. Recently, an improved method that replaced the p38 inhibitor (SB202190), as used in the original protocol, with IGF-1 and FGF-2 was reported by Fujii et al. [22]. As a result, we introduced an improved method to confirm cell proliferation and differentiation (Appendix A). With the improved culture conditions using IGF-1 and FGF-2, we tested whether Th2 cytokine stimulation promotes cell differentiation, as previously reported [12]. RNA-seq analysis and RT-PCR showed that IL-4 and IL-13 strongly induced tuft and goblet cell marker gene expression. Marker genes for Paneth cells, such as lysozyme (*LYZ*) and defensin alpha 6 (*DEFA6*), were also increased, while the stem cell marker gene *LGR5* was downregulated. However, it remains unclear whether IL-4 and IL-13 have positive effects on enterocytes and endocrine differentiation.

The data obtained using macaque organoids in the present study clearly support previous reports that tuft cells and goblet cells are induced by IL-4 and IL-13 [5,7,8]. In mouse studies, these Th2 cytokines are secreted from ILC2 in response to IL-25 produced by tuft cells and form tuft cell-ILC2 circuits [3,5,16]. In the present study, however, we failed to detect IL-25 transcripts in organoids treated with IL-4 or IL-13. Therefore, whether intestinal tuft cells are a major source of IL-25 in primate intestinal epithelia remains unclear. A recent study found that cysteinyl leukotrienes (cysLTs) synthesized by ALOX5 induced ILC2 activation, as well as IL-25 [23]. In our experiments, ALOX5 mRNA was increased in the IL-4-induced organoids. This indicates the possibility that cysLTs are utilized as drivers of type 2 immune responses in the macaque intestine. Notably, we failed to detect GNAT3, a tuft cell marker, in organoids stimulated with IL-4 or not, suggesting that GNAT3 is not a major component of tuft cells in macaques. Further studies are needed to unveil type 2 immunity system in the gut including interaction of IL-4/13 with other type 2 cytokines such as IL-31 and IL-33 [24].

There are some notable discrepancies between the results of previous studies using rodents and our study using macaques. In the present study, using macaque organoids, we observed the upregulation of genes selectively expressed in Paneth cells upon organoid stimulation by IL-4. On the other hand, using mice organoids, it has been reported that IL-4 induction suppresses gene expression in Paneth cells [25]. These differences suggest that the responses to cytokines differ between rodents and primates.

By quantifying ACh using HPLC/MS, we observed the accumulation of ACh in the organoids induced by IL-4. As most of the ChAT-immunoreactive tuft cells detected within organoids are tuft cells (Figure 3b), it is natural to speculate that ACh accumulation by IL-4 induction occurs within tuft cells. The question remains as to what the role of ACh is in tuft cells. It has also been reported that ACh secreted from tracheal tuft cells regulates mucociliary clearance in the airway [11,26,27]. We demonstrated for the first time in vitro that ACh triggers the release of antimicrobial peptides (AMPs) granules from Paneth cells in mouse intestinal organoids. This result was consistent with a previous report that Paneth cells are regulated by cholinergic stimuli [20,28]. Therefore, our findings suggest that ACh produced by tuft cells play a role in the secretion of granules from Paneth cells through paracrine stimulation (Figure 4c). The mechanism by which intestinal tuft cells secrete ACh is not well understood because of the missing link between ChAT-producing tuft cells and ACh vesicular transporters in the intestine. In general, the vesicular acetylcholine transporter (VAChT) is present where ChAT is expressed to load ACh in presynaptic vesicles [29]. The fact that no enrichment of VAChT mRNAs in the IL-4-induced organoids was observed suggests that intestinal tuft cells accumulate and secrete ACh by a molecular mechanism that differs from the conventional method using VAChT. However, further studies are needed to identify and confirm the regulatory mechanism of ACh secretion by intestinal tuft cells.

Paneth cells release antibacterial substances, such as defensins and lysozyme, to protect against invasion of foreign substances [28,30]. This phenomenon was observed in mouse intestinal organoids when LPS or bacteria were exposed to intestinal organoids [20]. Other studies have suggested that ACh is provided from vagal nerves so that local neurons also have the potential to stimulate Paneth cells [31].

Overall, we concluded that ACh production and accumulation are promoted by Th2 cytokines in macaque tuft cells. Our data provide useful insights into the function of ACh evoked by the immune response of tuft cells.

## 4. Materials and Methods

### 4.1. Animals

Three macaques (*Macaca mulatta* and *Macaca fuscata*, 0–6 years old) were used in the experiments. The study was approved by the Animal Welfare and Animal Care Committee of the Primate Research Institute, Kyoto University (permit numbers 2018-006 (1 April 2018), 2019-039 (1 April 2019), 2020-R2-A18 (1 April 2020)).

Two wild type (WT) C57BL/6 mice were purchased from CLEA Japan Inc. (Tokyo, Japan), housed and cared for under the “Guiding Principles in the Care and Use of Animals,” published by the Animal Care Committee of Tokyo University of Agriculture (#300111, 13 December 2018).

### 4.2. Preparation of Organoid Culture Media

For the organoid culture, Wnt3a-conditioned medium (Wnt3a-CM) was prepared using L1 cells and R-Spondion2-CM was prepared using HEK293T cells, as previously described [32,33]. The culture medium was prepared as follows. The basal medium and conventional proliferation medium, including SB202190 (FUJIFILM WAKO Pure Chemical Corporation, Osaka, Japan), were prepared as previously described [12]. Because an improved medium, with IGF-1 and FGF-2 instead of a p38 MAPK inhibitor (p38i) [22], was used for human organoid culture, we tested whether the improved media could be used in macaque organoids. The modified proliferation medium (IF condition) was prepared by supplementing the basal medium with 50% Wnt3a-CM, 10% R-Spondin2-CM, 0.5% (*w*/*v*) Albumax I Lipid-Rich BSA (Thermo Fisher Scientific, Waltham, MA, USA), 100 ng/mL recombinant human Noggin (Proteintech, Rosemont, IL, USA), 100 ng/mL recombinant human IGF-1 (FUJIFILM WAKO Pure Chemical Corporation), 50 ng/mL recombinant human FGF-basic (FGF-2) (FUJIFILM WAKO Pure Chemical Corporation), 500 nM A83-01 (FUJIFILM WAKO Pure Chemical Corporation), 1 mM N-acetylcysteine (Sigma-Aldrich, St. Louis, MO, USA), 10 nM human [Leu^15^]-gastrin 1 (Sigma-Aldrich), 10 mM nicotinamide (FUJIFILM WAKO Pure Chemical Corporation), and 50 ng/mL recombinant human EGF (Sigma-Aldrich). The differentiation medium (DI condition) was prepared by removing nicotinamide and EGF from IF condition. For organoid differentiation into tuft cells, 400 ng/mL IL-4 or 40 ng/mL IL-13 (PeproTech, Cranbury, NJ, USA) were added to the DI condition (IL-4 or IL-13).

### 4.3. Crypt Isolation and Organoid Culture from Macaques

Crypt isolation from the small intestinal samples (jejunum) and crypt embedding in Matrigel (CORNING, Corning, NY, USA) were performed as previously described [12]. After Matrigel polymerization, crypts were covered with medium of the IF condition. The medium was supplemented with 10 mM Y-27632 (Nacalai tesque, Kyoto, Japan) for the first three days of culture. The medium was replaced with fresh medium every two to three days. The organoids were passaged every six to nine days using TrypLE Express (Thermo Fisher Scientific) for dissociation into a single cell. In the differentiation culture of organoids, the organoids were cultured in the DI condition or IL-4 or IL-13 condition for 72 h after growth under the IF condition for six to nine days.

### 4.4. Immunostaining of Tissues and Organoids

Frozen tissue sections were prepared as follows. The intestinal samples obtained from macaques were shredded into small pieces and fixed with 4% paraformaldehyde for 1 h at 4 °C. Next, the tissue samples were replaced in 30% (*w*/*v*) sucrose (Nacalai tesque) in PBS at 4 °C and embedded in Tissue-Tek O.C.T Compound (Sakura Finetek Japan, Tokyo, Japan). Then, cryosections with a thickness of 12 μm were prepared using a CM1850 cryostat (Leica Microsystems, Wetzlar, Germany). The organoid samples were prepared as previously described [12], with some modifications. Briefly, the organoids were harvested in a 1.5 mL microtube by replacing Matrigel with cold DPBS. The organoids were fixed with 4% paraformaldehyde for 10–20 min and washed with DPBS.

Immunostaining was performed as follows. After incubation with blocking buffer (0.3% (*v*/*v*) Triton X-100 and 2% (*w*/*v*) normal donkey serum in DPBS) for 1 h at room temperature (for tissues) or overnight at 4 °C (for organoids), and then incubation with the primary antibodies described below overnight at 4 °C, primary antibodies were detected using 488 or 555 Alexa secondary antibodies (1:1000) (Thermo Fisher Scientific) for 1 h at room temperature (for tissues) or overnight at 4 °C (for organoids). Counterstaining of nuclei was performed using DAPI solution (1:1000) (FUJIFILM WAKO Pure Chemical Corporation). The following primary antibodies were used: rabbit anti-DCAMKL1 (DCLK1) (1:400) (abcam, Cambridge, UK), goat anti-5HT (1:1000) (Immunostar, Hudson, WI, USA), rabbit anti-MUC2 (1:400) (abcam), rabbit anti-DEFA6 (1:400) (Sigma-Aldrich), and goat anti-ChAT (1:400) (Sigma-Aldrich). Fluorescence images of the organoids were acquired using a confocal scanning microscope (FV-1200; Olympus, Tokyo, Japan). The TLC frequency was calculated from the Z-axis scanning images of the organoids.

### 4.5. RNA-Seq Analysis

Matrigel around the organoids was removed by washing with cold DPBS, and total RNA was purified using ISOGEN (Nippon Gene, Tokyo, Japan) or RNeasy Mini Kit (Qiagen, Hilden, Germany), according to the manufacturer’s instructions. Its quality was confirmed using a Bioanalyzer (Agilent Technologies, Santa Clara, CA, USA).

RNA-seq of the organoids induced by IL-4 and IL-13 was performed as follows. The mRNA was isolated from approximately 300 ng of total RNA using the Dynabeads mRNA Purification Kit.

The sequencing library was prepared using the MGIEasy RNA Directional Library Prep Kit. Next, single-end sequencing was performed using the MGISEQ-2000 (DNBSEQ-G400) sequencing instrument, according to the manufacturer’s instructions, yielding 50 bp single-end reads. Base calling and the generation of FastQ files on the DNBSEQ-G400 was performed using Zebracall (version 1.0.7) provided by the manufacturer [34]. The read data were deposited in the DNA Data Bank of the Japan Sequence Read Archive (accession numbers GSE178817 and DRA011980).

The software trim_galore (https://www.bioinformatics.babraham.ac.uk/projects/trim_galore/, version 0.6.4_dev, accessed on 1 August 2019) was used for sequencing sequence quality control and adapter trimming. Gene expression was evaluated using fast aligner kallisto (https://pachterlab.github.io/kallisto/, version 0.45.0, accessed on 11 May 2019). A sequence database of Macaca mRNA was constructed using Macaca_mulatta.Mmul_8.0.1.cdna.abinitio.fa file retrieved from ENSEMBL.

Differentially expressed genes were identified using DESeq2 package (http://bioconductor.org/packages/release/bioc/html/DESeq2.html, version 1.26.0, accessed on 29 October 2019).

Clustering of the gene expression pattern of each group was visualized as a principal component analysis (PCA) using the R “prcomp” package and a heatmap using the R “pheatmap” package. Differentially expressed genes were identified using DESeq2 package (http://bioconductor.org/packages/release/bioc/html/DESeq2.html, version 1.26.0, accessed on 29 October 2019).

### 4.6. Semi-Quantitative Reverse Transcription-Polymerase Chain Reaction

The cDNA library was synthesized from 500 ng of total RNA (measured by NanoDrop UV spectrophotometer; Thermo Fisher Scientific) using the SuperScript III First-Strand Synthesis System (Thermo Fisher Scientific) and used as a template for RT-PCR using GoTaq polymerase (Promega, Madison, WI, USA) according to the manufacturer’s instructions. The primers were designed based on the mRNA sequences of *Macaca mulatta* (Appendix A). The products were electrophoresed on a 1.5% agarose gel and stained with ethidium bromide. The intensity of the bands was calculated using Image J (NIH) and normalized to G3PDH expression.

### 4.7. Preparation of the Organoid Samples for the Quantification of Acetylcholine Using HPLC/MS

Macaque organoids cultured with IF, DI, and IL-4 medium were collected after the differentiation culture and then washed with DPBS to remove the Matrigel completely. The organoids were then homogenized in TNE buffer (20 mM Tris-HCl, 150 mM NaCl, and 1 mM EDTA) with 0.1% Triton X-100 using a 23G needle. After centrifugation for 10 min at 15,000 rpm at 4 °C, the supernatants were collected and used as samples for HPLC/MS analysis.

### 4.8. Calibration Standard Preparation for HPLC/MS Analysis

Acetylcholine chloride (ACh-Cl) (Sigma-Aldrich, St. Louis, MO, USA) was used as a standard sample and acetylcholine-d4 chloride (d4-ACh) (FUJIFILM WAKO Pure Chemical) was used as an internal standard (IS) for calibration. An aqueous solution containing 1000 μg/mL of ACh-Cl or ACh-d4 was prepared as a stock solution. A series of 0.14, 0.41, 1.23, 3.7, 11.1, 33.3, and 100 ng/mL standard samples containing 1 ng/mL ACh-d4 were prepared from the stock solution. Each solution (100 μL) was placed into a 1.5 mL sample tube, to which methanol (250 μL) and chloroform (200 μL) were added. The resulting mixture was vortexed for 1 min and centrifuged at 15,000 rpm for 10 min at 4 °C. After centrifugation, 180 μL of the supernatant was collected in an autosampler vial, 20 μL of which was injected into the LC–MS/MS system. The calibration curve was generated by plotting the nominal concentrations of ACh against the peak area ratio of ACh-Cl to IS.

### 4.9. Pretreatment of Organoid Samples for HPLC/MS Analysis

A 50 μL aliquot of sample and 50 μL of IS solution (2 ng/mL) were placed into a 1.5 mL sample tube, to which methanol (250 μL) and chloroform (200 μL) were added for deproteinization. The obtained mixture was vortexed for 1 min and centrifuged at 15,000 rpm for 10 min at 4 °C. After centrifugation, 180 μL of the supernatant was collected in an autosampler vial, of which 20 μL was injected into the LC–MS/MS system.

### 4.10. Quantification of Acetylcholine Using HPLC/MS

LC-MS analyses were performed as follows. The resulting supernatant was analyzed using an HPLC-Orbitrap apparatus (Thermo Fisher Scientific). The HPLC conditions were as follows: Thermo Fisher Scientific Ultimate 3000; column: Imtakt Scherzo SS C-18 (150 mm × 5 mm, 3 µm, Imtakt Corporation, Kyoto, Japan); column oven: 30 °C; mobile phase (0.5% formic acid/45 mM ammonium formate–acetonitrile (65:35, v/v)): 0 min (90/10), 30–40 min (0/100), 41–60 min (90/10); flow rate, 0.4 mL/min. The Orbitrap conditions were as follows: Thermo Fisher Scientific Q Exactive Focus Orbitrap LC-MS/MS System; ionization mode: ESI (positive); measurement mode: SIM (selected ion monitoring, *m*/*z*: 146.11756 and 150.14266); heater temperature: 200 °C; spray voltage: 3.5 kV; capillary temperature: 350 °C; sheath gas: 50 L/h; auxiliary gas: 15 L/h; collision energy: 35 eV.

### 4.11. Imaging of Granule Secretion from Paneth Cells Using Organoids

Imaging of granule secretion was performed using mouse ileum organoids on day 3. The generation and culture of mouse intestinal organoids were performed as previously described [35]. The organoids suspended with Matrigel were transferred thinly onto a glass bottom dish (Greiner Bio-One). The dish was incubated on ice for 5 min to allow the organoids to drop to the bottom. Then, Matrigel was polymerized at 37 °C in a CO_2_ incubator. After polymerization, the organoids in Matrigel were covered with advanced Dulbecco’s modified Eagle medium/Ham’s F12 until analysis.

The DIC images of the Paneth cells before and after stimulation were acquired using a Leica DMI6000B microscope (Leica Microsystems) at 16 frame/s. The area of granules in Paneth cells was measured from differential interference contrast images before and after stimulation using Image J (NIH).

### 4.12. Statistical Analysis

All statistical analysis of data was performed on R packages. Tukey’s honest significant difference (Tukey’s HSD) test was used for the pairwise comparisons of the data, and statistically significant differences were found by one-way analysis of variance (ANOVA) (*p* < 0.05).

## Figures and Tables

**Figure 1 ijms-22-07921-f001:**
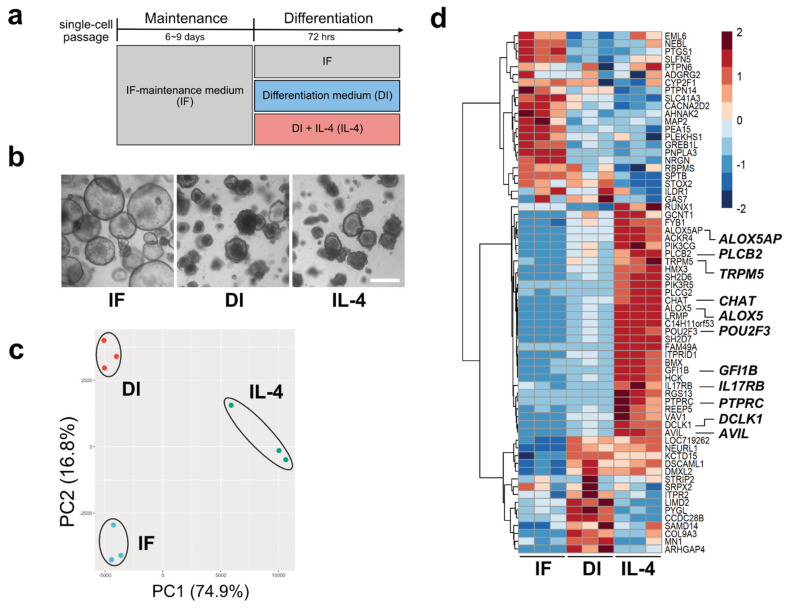
Tuft cell induction and verification. (**a**). The strategy of inducing tuft cells by changing culture media. (**b**) Representative images of the intestinal organoids cultured in three different media for 72 h. The morphology of the organoids cultured in differentiation (DI) or interleukin (IL)-4 media changed, becoming more shrunken or budding in shape, while organoids cultured under the IF media maintained a ball-like structure. Scale bar, 500 µm. (**c**). Principal component analysis (PCA) for the transcriptome profiles of the organoids cultured in control (IF), DI, and IL-4 medium. The scatter plots based on PCA scores of PC1 and PC2 represent the similarity among each biological replicate in three groups. (**d**). A heatmap of 67 DEGs for the tuft cell in the organoids cultured in IF, DI, or IL-4 medium. Colors indicate the relative expression levels based on row-wise z-score of transformed TPM values.

**Figure 2 ijms-22-07921-f002:**
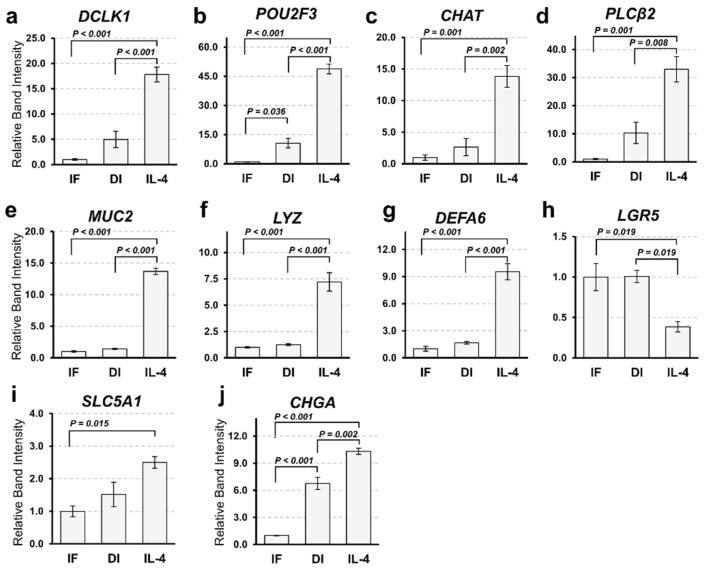
Gene expression analysis of intestinal markers by semi-quantitative reverse transcription polymerase chain reaction (RT-PCR). (**a**–**j**). Comparison of the expression marker genes under three different culture conditions. The mRNA expression of organoids cultured in IF, DI, or IL-4 media was subjected to semi-quantitative analysis. Marker genes of tuft cells (**a**–**c**), goblet cells (**e**), Paneth cells (**f**–**g**), intestinal stem cells (**h**), enterocytes (**i**), and enteroendocrine cells (**j**) were amplified by RT-PCR using gene specific primers. The expression level of each gene was examined by band intensity after electrophoresis. Data are shown as means ± SEM (*N* = 3); *p*-values are shown in each figure, Tukey’s honest significant difference (HSD), analysis of variance (ANOVA) *p* < 0.001 (**a**–**h**), *p* = 0.016 (**i**), and *p* = 0.012 (**j**).

**Figure 3 ijms-22-07921-f003:**
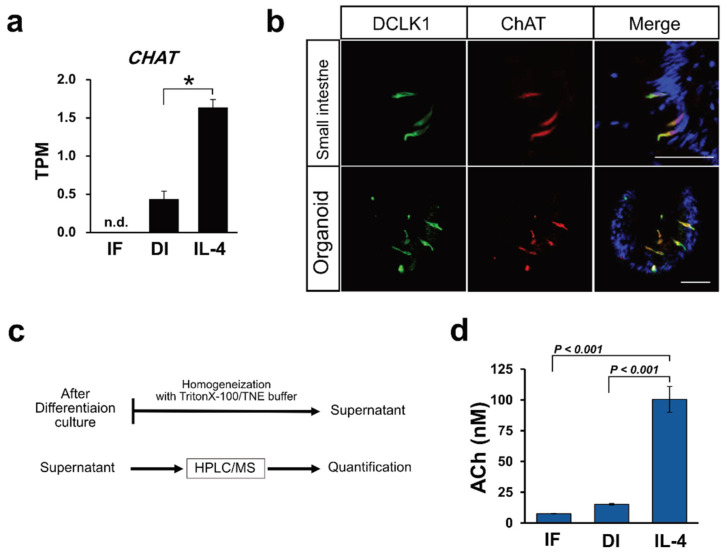
Acetylcholine production correlates with tuft cell differentiation. (**a**) CHAT mRNAs were induced by IL-4 induction. Data are shown as means ± SEM (*N* = 3); * adjusted *p*-value < 0.0001 vs. DI group, R package “DESeq2”. (**b**) Immunohistochemical analysis of tissues from the small intestine and from macaque organoids show that DCLK1 positive cells (green) and ChAT positive cells (red) were colocalized. Nuclei were stained with DAPI. Scale bar, 50 µm. (**c**). A schematic flow chart of ACh quantification. (**d**) Quantification of Ach in the organoids cultured in IF, DI, or IL-4 medium. Samples obtained from the organoids were analyzed by HPLC/MS and ACh level in each sample was calculated. Data are shown as means ± SEM (*N* = 3); *p*-values are shown in each figure, Tukey’s HSD.

**Figure 4 ijms-22-07921-f004:**
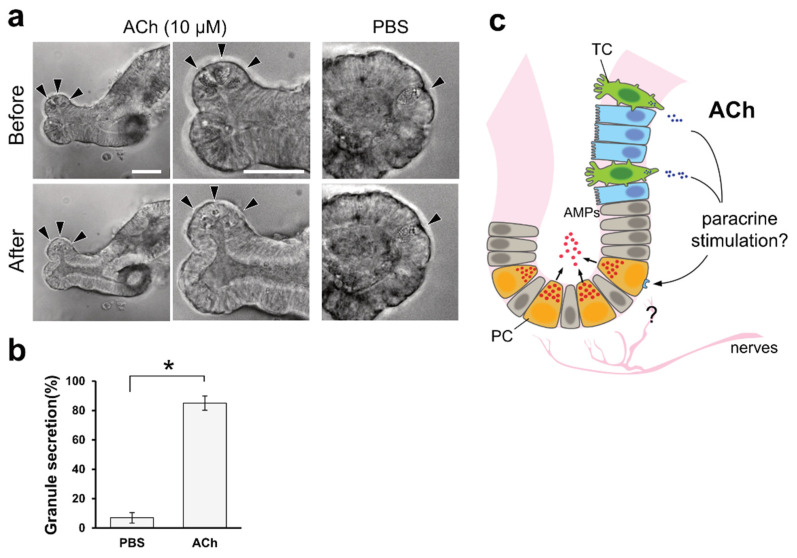
ACh induces granular secretion from Paneth cells in mouse intestinal organoids. (**a**). Representative images of Paneth cells (arrowheads) before and after 5 min of 10 µM ACh stimulation. PBS was used as a negative control. Granules in the apical part of the Paneth cells were released inside the organoid upon ACh stimuli, and very few granules were observed. Scale bar, 100 µm. (**b**). Granular secretion was statistically measured by calculating the area of granules, as described in the Materials and Methods. The secretion ratio was described as the ratio of the post-stimulation area to the pre-stimulation granule area. * *p* < 0.001, Welch’s *t*-test. (**c**). Hypothetical model derived from this study. ACh is basolaterally secreted from tuft cells and acts as a paracrine stimulus on Paneth cells to secrete granules that contain antimicrobial peptides in the intestine. TC, PC, and AMPs represent tuft cells, Paneth cells, and antimicrobial peptides, respectively.

## Data Availability

All the data are shown in the main manuscript and in the Appendix A.

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
