# Peer review of "Interleukin-4 Promotes Tuft Cell Differentiation and Acetylcholine Production in Intestinal Organoids of Non-Human Primate"

_ijms, 2021, doi:10.3390/ijms22157921_

Round 1
Reviewer 1 Report
The manuscript is interesting and well written. However, I suggest ot discuss the relation between IL-33 and IL-4 and between vitamin D and IL-4 (see and add as references paper by Murdaca et al concerning IL-33/31 axis and concerning vitamin D and immune system)
Author Response
We are most grateful to the reviewer for the helpful comments on our manuscript. As suggested by the reviewer, we have discussed about relations of IL-31 and IL-33 with IL-4/13 with a reference of Murdaca et al (24) in the manuscript.
All the changes we made are marked in Yellow including changes made to reply to other Reviewers.
We are grateful to the reviewer for the time and effort with our manuscript.
Reviewer 2 Report
In this research article the authors isolated crypt cells from the small intestine of primates to produce organoids that were subsequently differentiated into tuft cells. For that reason, they treated the organoids with Th2 cytokines and recorded changes in the expression of tuft, Paneth, and Goblet cell markers. Furthermore, IL-4 and -13 treatments induced global transcriptomic changes in the cells, while IL-4 promoted the accumulation of acetylcholine and excretion of granules from Paneth cells.
Materials and methods are carefully documented, and the procedures seem appropriate for testing the author’s hypothesis.
Some points that would require attention are:
Abstract:
Lines 13-14: These are well-known facts, could be omitted or edited.
Lines 16-17: This reference seems out of place.
Lines 18-19: do the authors mean that their function in the gut remains poorly understood?
Line 19: in this work no genetic manipulation was performed by the authors, the cellular events that were recorded were due to the activation of cellular cascades by IL-4, -13, and thus changes in the expression of genes and not of their sequence.
Ιntroduction:
Line 34: the word “rare” could be replaced.
Lines 39-41: are tuft cells only involved in anti-parasite responses? What are other physiological functions that they execute in the gut?
Line 46: “synthetic enzymes” could be replaced by “biosynthetic enzymes” or “enzymes for the biosynthesis of…”
Results
Line 66: “media modified from the original composition” could be replaced with “modified media”.
Lines 70-72: how did the authors determine the proliferation rate of the cells? Are there any supplementary graphs that would support this claim?
Lines 74-77: appropriate citations should be provided for the correlation of specific cell markers with cell types.
Line 104-105: GNAT3 and IL-25 are tuft cell markers in what organism? The authors should include this information in the text.
Methods
The authors could add the location of the companies, consumables were purchased from.
Lines 253-254: are the protocols for in-house production of Wnt3a- and R-Spondin- conditioned media previously published? If so, the authors should include the appropriate citations.
Lines 259-260: The authors should rephrase these lines about the composition of DI condition medium. Nicotinamide and EGF were not removed from IF, they were not added in the first place.
Lines 304-319: are these tools previously published from the groups that developed them? If yes, please add appropriate citations in the text.
Lines 320-327: how was RNA concentration determined?
Lines 328-335: what was the rationale for using only IL-4 and not IL-13 for the differentiation of organoids into tuft cells? The authors could add a comment wherever they find fitting.
Author Response
We thank to the reviewer for the helpful comments and have changed our manuscript as has been suggested. The helpful comments have led to a stronger manuscript.
All the changes we made are marked in Yellow including changes made to reply to other Reviewers.
Lines 13-14: These are well-known facts, could be omitted or edited.
We have omitted the first sentence of the abstract as the reviewer has suggested.
Lines 16-17: This reference seems out of place.
We have omitted the sentence in the lines 16-17.
Lines 18-19: do the authors mean that their function in the gut remains poorly understood?
Yes. We have inserted the phrase "in the gut" within the sentence.
Line 19: in this work no genetic manipulation was performed by the authors, the cellular events that were recorded were due to the activation of cellular cascades by IL-4, -13, and thus changes in the expression of genes and not of their sequence.
We appreciate the reviewer for pointing this out. We have edited the sentence to "we investigated changes in the expression of genes.....". (Line 18)
Ιntroduction:
Line 34: the word “rare” could be replaced.
We have replaced to "an atypical".
Lines 39-41: are tuft cells only involved in anti-parasite responses? What are other physiological functions that they execute in the gut?
We have added several examples of tuft cell function in the gut with references as follows. "Other physiological functions of tuft cells in the gut are also proposed for functions such as to sense bitter compounds and succinate that will guide hyperplasia of tuft cells (6-9)." (Line 43-45)
Line 46: “synthetic enzymes” could be replaced by “biosynthetic enzymes” or “enzymes for the biosynthesis of…”
We have replaced “synthetic enzymes” to "biosynthetic enzymes" as the reviewer suggested.
Results
Line 66: “media modified from the original composition” could be replaced with “modified media”.
We have replaced to "modified media".
Lines 70-72: how did the authors determine the proliferation rate of the cells? Are there any supplementary graphs that would support this claim?
Yes, we have added a figure (Supplementary Figure 1b) to show that modified media could support more organoids with larger size. Accordingly we have edited the manuscript. (Lines 71-73)
Lines 74-77: appropriate citations should be provided for the correlation of specific cell markers with cell types.
We have added three citations referring specific cell markers. (Lines 76-78)
Line 104-105: GNAT3 and IL-25 are tuft cell markers in what organism? The authors should include this information in the text.
GNAT3 and IL25 are tuft cell markers at least in mice. Therefore, we have added the phrase "in mouse" to the sentence which the reviewer pointed out. We also added the reference. (Line 109)
Methods
The authors could add the location of the companies, consumables were purchased from.
Yes, we have added necessary informations for the reagents and equipments in Materials and Methods.
Lines 253-254: are the protocols for in-house production of Wnt3a- and R-Spondin- conditioned media previously published? If so, the authors should include the appropriate citations.
We have cited the work which uses Wnt3a- and R-Spondin-conditioned medium in the revised manuscript (reference 32, 33). Since there are three R-Spondin family, we specified that the R-Spondin we used in our experiment was R-Spondin2. (Lines 254-256)
Lines 259-260: The authors should rephrase these lines about the composition of DI condition medium. Nicotinamide and EGF were not removed from IF, they were not added in the first place.
We have described IF condition that include nicotinamide and EGF, and then described about DI condition medium. We have added a word “condition” after IF in the Line 271.
Lines 304-319: are these tools previously published from the groups that developed them? If yes, please add appropriate citations in the text.
Yes, we have added one citation that readers could follow the method. The paper we have cited is: Huang J et al. A reference human genome dataset of the BGISEQ-500 sequencer. Gigascience. 2017;6:1–9. The reference number for the paper is 34. (Line 319)
Lines 320-327: how was RNA concentration determined?
RNA concentration was measured by NanoDrop UV spectrometer (Thermo Fisher Scientific). We have included this information in Materials and Methods.
Lines 328-335: what was the rationale for using only IL-4 and not IL-13 for the differentiation of organoids into tuft cells? The authors could add a comment wherever they find fitting.
We have also used IL-13 and observed similar differentiation event. We confirmed this by RNA-Seq analysis as well as immunostaining. Therefore, in the result section we wrote: "To substantiate transcriptional regulation by Th2 cytokines, IL-13 was also added to the culture medium, and the expression of tuft cell marker proteins and transcripts was analyzed by immunohistochemistry and RNA-Seq. The results showed that both protein and gene expression patterns were similar to those stimulated by IL-4, indicating that IL-13 also induces tuft cell differentiation (Figure S3a, b).” (Lines 110-114).
Reviewer 3 Report
This manuscript demonstrates upregulation of ACh 25 by IL-4 induction in primates, suggesting that IL-4 plays a role in the secretion of Paneth cell granules 26 through paracrine stimulation.
This is a very complete and interesting work of great utility for the scientific community.
After reading it I have some questions for and concerns for the authors:
- Please review the definition of abbreviations throughout the text. Some are missing. Also, define those used in each figure.
- It is not clear to me if the modified medium used in these experiments is the same as the one named in the discussion and referenced as 14. Is the only difference the origin (human or macaque)?
- Have the authors considered assessing possible changes in permeability, e.g. through the expression of occludin?
Author Response
We are grateful to the reviewer for the helpful comments. We have tried to answer each comment that have led to a stronger manuscript. Please see changes marked with Yellow in the manuscript and our response to each comment as follows.
- Please review the definition of abbreviations throughout the text. Some are missing. Also, define those used in each figure.
We have added the definition of the abbreviations in the manuscript as well as in figures.
- It is not clear to me if the modified medium used in these experiments is the same as the one named in the discussion and referenced as 14. Is the only difference the origin (human or macaque)?
Thank you for the comment. Yes, since the modified medium is new to us using macaque organoids, we tried show that the modified medium could be also applied to the macaque intestinal organoid culture. In the Result section, we wrote “We tested whether modified media could be used to culture macaque intestinal organoids.” (Line 68-69)
- Have the authors considered assessing possible changes in permeability, e.g. through the expression of occludin?
I appreciate the suggestion for the future project. We are going to examine whether permeability changes in the next step, but not in this paper.
We thank the reviewer for the time and effort with our manuscript.
Reviewer 4 Report
- The authors explored the topic and they obtained the purpose of the study. This research is the first to demonstrate ACh 25 upregulation by IL-4 induction in primates, suggesting that IL-4 plays a role in Paneth cell granule secretion via paracrine stimulation.
- The paper is well written and text is clear to read.
- The methods used are sufficiently documented. Results obtained are well explained and data interpretation is also correct. Conclusions are consistent with the evidence and arguments presented.
- About limitations the authors should modify and standardize statistical analysis.
-
Specific indications:
- The following acronyms should be reported in extenso at their first appearance in the text: DCLK1; MUC2; TFF3; DEFA6; LYZ; DEFA6; ALOX5; POUF3; PLCB2; PLCG2; IL17RB; ALOX5.
- In the legend of figures 2-4 the authors should add the ANOVA P value;
- In the legend of figures the authors should add informations about *P values Vs. ...group;
- The authors should include the number of animals in the Materials and Methods section;
- The authors should add the paragraph Statistical Analysis in the section Materials and Methods.
Author Response
We appreciate the reviewer for the comments that will improve our manuscript. Please see our response to the reviewer as follows and also please see changes in the manuscript which are marked in Yellow including all the changes.
- The following acronyms should be reported in extenso at their first appearance in the text: DCLK1; MUC2; TFF3; DEFA6; LYZ; DEFA6; ALOX5; POUF3; PLCB2; PLCG2; IL17RB; ALOX5.
We have added the original term before abbreviation.
- In the legend of figures 2-4 the authors should add the ANOVA P value;
We very much appreciate the reviewer for the comments on statistics. According to the comments, we have added P values where needed.
- In the legend of figures the authors should add informations about *P values Vs. ...group;
We have adde *P values and Vs. ...group. Please see changes in the legend and figures as well in the revised manuscript.
- The authors should include the number of animals in the Materials and Methods section;
According to the comment, we have added the numbers of animals (three macaques and two mice) we used for the experiment. (Lines 246 and 250)
- The authors should add the paragraph Statistical Analysis in the section Materials and Methods.
We appreciate for the comment. We have added a paragraph for Statistic Analysis to Materials and Methods. (Lines 392-395)